# Inducing brain-relevant bias
# in natural language processing models

**Dan Schwartz**
Carnegie Mellon University
drschwar@cs.cmu.edu

**Mariya Toneva**
Carnegie Mellon University
mariya@cmu.edu

**Leila Wehbe**
Carnegie Mellon University
lwehbe@cmu.edu

## Abstract

Progress in natural language processing (NLP) models that estimate representations of word sequences has recently been leveraged to improve the understanding of language processing in the brain. However, these models have not been specifically designed to capture the way the brain represents language meaning. We hypothesize that fine-tuning these models to predict recordings of brain activity of people reading text will lead to representations that encode more brain-activity-relevant language information. We demonstrate that a version of BERT, a recently introduced and powerful language model, can improve the prediction of brain activity after fine-tuning. We show that the relationship between language and brain activity learned by BERT during this fine-tuning transfers across multiple participants. We also show that, for some participants, the fine-tuned representations learned from both magnetoencephalography (MEG) and functional magnetic resonance imaging (fMRI) are better for predicting fMRI than the representations learned from fMRI alone, indicating that the learned representations capture brain-activity-relevant information that is not simply an artifact of the modality. While changes to language representations help the model predict brain activity, they also do not harm the model's ability to perform downstream NLP tasks. Our findings are notable for research on language understanding in the brain.

## 1   Introduction

The recent successes of self-supervised natural language processing (NLP) models have inspired researchers who study how people process and understand language to look to these NLP models for rich representations of language meaning. In these works, researchers present language stimuli to participants (e.g. reading a chapter of a book word-by-word or listening to a story) while recording their brain activity with neuroimaging devices (fMRI, MEG, or EEG), and model the recorded brain activity using representations extracted from NLP models for the corresponding text. While this approach has opened exciting avenues in understanding the processing of longer word sequences and context, having NLP models that are specifically designed to capture the way the brain represents language meaning may lead to even more insight. We posit that we can introduce a brain-relevant language bias in an NLP model by explicitly training the NLP model to predict language-induced brain recordings.

In this study we propose that a pretrained language model — BERT by Devlin *et al.* (2018) — which is then fine-tuned to predict brain activity will modify its language representations to better encode the information that is relevant for the prediction of brain activity. We further propose fine-tuning

Code available at https://github.com/danrsc/bert_brain_neurips_2019

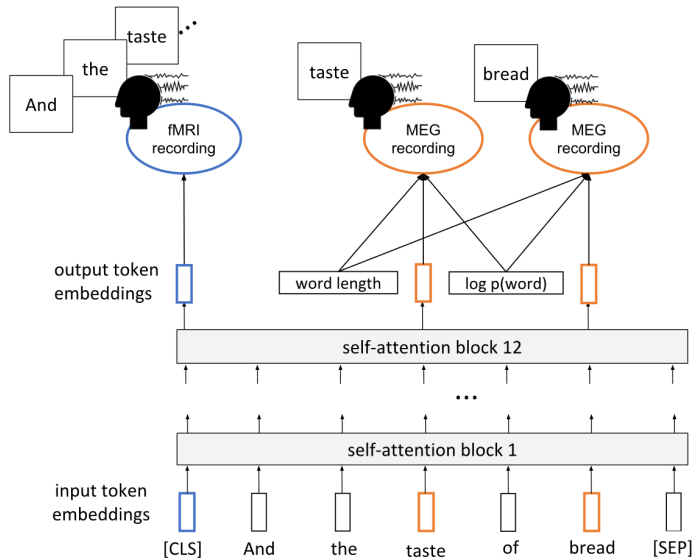

Figure 1: General approach for fine-tuning BERT using fMRI and/or MEG data. A linear layer maps the output token embeddings from the base architecture to brain activity recordings. Only MEG recordings that correspond to content words in the input sequence are considered. We include the word length and context-independent log-probability of each word when predicting MEG. fMRI data are predicted from the pooled embedding of the sequence, i.e. the [CLS] token embedding. For more details of the procedure, see section 3.2.

simultaneously from multiple experiment participants and multiple brain activity recording modalities to bias towards representations that generalize across people and recording types. We suggest that this fine-tuning can leverage advances in the NLP community while also considering data from brain activity recordings, and thus can lead to advances in our understanding of language processing in the brain.

## 2   Related Work

The relationship between language-related brain activity and computational models of natural language (NLP models) has long been a topic of interest to researchers. Multiple researchers have used vector-space representations of words, sentences, and stories taken from off-the-shelf NLP models and investigated how these vectors correspond to fMRI or MEG recordings of brain activity (Mitchell *et al.*, 2008; Murphy *et al.*, 2012; Wehbe *et al.*, 2014b,a; Huth *et al.*, 2016; Jain and Huth, 2018; Pereira *et al.*, 2018). However, few examples of researchers using brain activity to modify language representations exist. Fyshe *et al.* (2014) builds a non-negative sparse embedding for individual words by constraining the embedding to also predict brain activity well, and Schwartz and Mitchell (2019) very recently have published an approach similar to ours for predicting EEG data, but most approaches combining NLP models and brain activity do not modify language embeddings to predict brain data. In Schwartz and Mitchell (2019), the authors predict multiple EEG signals on a dataset using a deep network, but they do not investigate whether the model can transfer its representations to new experiment participants or other modalities of brain activity recordings.

Because fMRI and MEG/EEG have complementary strengths (high spatial resolution vs. high temporal resolution) there exists a lot of interest in devising learning algorithms that combine both types of data. One way that fMRI and MEG/EEG have been used together is by using fMRI for better source localization of the MEG/EEG signal (He *et al.*, 2018) (source localization refers to inferring the sources in the brain of the MEG/EEG recorded on the head). Palatucci (2011) uses CCA to map between MEG and fMRI recordings for the same word. Mapping the MEG data to the common space allows the authors to better decode the word identity than with MEG alone. Cichy *et al.* (2016) propose a way of combining fMRI and MEG data of the same stimuli by computing stimuli similarity matrices for different fMRI regions and MEG time points and finding corresponding regions and time points. Fu *et al.* (2017) proposes a way to estimate a latent space that is high-dimensional both in time and space from simulated fMRI and MEG activity. However, effectively combining fMRI and MEG/EEG remains an open research problem.

# 3 Methods

## 3.1 MEG and fMRI data

In this analysis, we use magnetoencephalography (MEG) and functional magnetic resonance imaging (fMRI) data recorded from people as they read a chapter from *Harry Potter and the Sorcerer's Stone* Rowling (1999). The MEG and fMRI experiments were shared respectively by the authors of Wehbe *et al.* (2014a) at our request and Wehbe *et al.* (2014b) online[1]. In both experiments the chapter was presented one word at a time, with each word appearing on a screen for 0.5 seconds. The chapter included 5176 words.

MEG was recorded from nine experiment participants using an Elekta Neuromag device (data for one participant had too many artifacts and was excluded, leaving 8 participants). This machine has 306 sensors distributed into 102 locations on the surface of the participant's head. The sampling frequency was 1kHz. The Signal Space Separation method (SSS) (Taulu *et al.*, 2004) was used to reduce noise, and it was followed by its temporal extension (tSSS) (Taulu and Simola, 2006). The signal in every sensor was downsampled into 25ms non-overlapping time bins, meaning that each word in our data is associated with a 306 sensor $\times$ 20 time points image.

The fMRI data of nine experiment participants were comprised of $3 \times 3 \times 3mm$ voxels. Data were slice-time and motion corrected using SPM8 (Kay *et al.*, 2008). The data were then detrended in time and spatially smoothed with a $3mm$ full-width-half-max kernel. The brain surface of each subject was reconstructed using Freesurfer (Fischl, 2012), and a thick grey matter mask was obtained to select the voxels with neuronal tissue. For each subject, 50000-60000 voxels were kept after this masking. We use Pycortex (Gao *et al.*, 2015) to handle and plot the fMRI data.

## 3.2 Model architecture

In our experiments, we build on the BERT architecture (Devlin *et al.*, 2018), a specialization of a transformer network (Vaswani *et al.*, 2017). Each block of layers in the network applies a transformation to its input embeddings by first applying self-attention (combining together the embeddings which are most similar to each other in several latent aspects). These combined embeddings are then further transformed to produce new features for the next block of layers. We use the PyTorch version of the BERT code provided by Hugging Face[2] with the pretrained weights provided by Devlin *et al.* (2018). This model includes 12 blocks of layers, and has been trained on the BooksCorpus (Zhu *et al.*, 2015) as well as Wikipedia to predict masked words in text and to classify whether two sequences of words are consecutive in text or not. Two special tokens are attached to each input sequence in the BERT architecture. The [SEP] token is used to signal the end of a sequence, and the [CLS] token is trained to be a sequence-level representation of the input using the consecutive-sequence classification task. Fine-tuned versions of this pretrained BERT model have achieved state of the art performance in several downstream NLP tasks, including the GLUE benchmark tasks (Wang *et al.*, 2018). The recommended procedure for fine-tuning BERT is to add a simple linear layer that maps the output embeddings from the base architecture to a prediction task of interest. With this linear layer included, the model is fine-tuned end-to-end, i.e. all of the parameters of the model change during fine-tuning. For the most part, we follow this recommended procedure in our experiments. One slight modification we make is that in addition to using the output layer of the base model, we also concatenate to this output layer the word length and context-independent log-probability of each word (see Figure 1). Both of these word properties are known to modulate behavioral data and brain activity (Rayner, 1998; Van Petten and Kutas, 1990). When a single word is broken into multiple word-pieces by the BERT tokenizer, we attach this information to the first token and use dummy values (0 for word length and -20 for the log probability) for the other tokens. We use these same dummy values for the special [CLS] and [SEP] tokens. Because the time-resoluton of fMRI images is too low to resolve single words, we use the pooled output of BERT to predict fMRI data. In the pretrained model, the pooled representation of a sequence is a transformed version of the embedding of the [CLS] token, which is passed through a hidden layer and then a tanh function. We find empirically that using the [CLS] output embedding directly worked better than using this transformation, so we use the [CLS] output embedding as our pooled embedding.

### 3.3 Procedure

**Input to the model.**   We are interested in modifying the pretrained BERT model to better capture brain-relevant language information. We approach this by training the model to predict both fMRI data and MEG data, each recorded (at different times from different participants) while experiment participants read a chapter of the same novel. fMRI records the blood-oxygenation-level dependent (BOLD) response, i.e. the relative amount of oxygenated blood in a given area of the brain, which is a function of how active the neurons are in that area of the brain. However, the BOLD response peaks 5 to 8 seconds after the activation of neurons in a region (Nishimoto *et al.*, 2011; Wehbe *et al.*, 2014b; Huth *et al.*, 2016). Because of this delay, we want a model which predicts brain activity to have access to the words that precede the timepoint at which the fMRI image is captured. Therefore, we use the 20 words (which cover the 10 seconds of time) leading up to each fMRI image as input to our model, irrespective of sentence boundaries. In contrast to the fMRI recordings, MEG recordings have much higher time resolution. For each word, we have 20 timepoints from 306 sensors. In our experiments where MEG data are used, the model makes a prediction for all of these $6120 = 306 \times 20$ values for each word. However, we only train and evaluate the model on content words. We define a content word as any word which is an adjective, adverb, auxiliary verb, noun, pronoun, proper noun, or verb (including to-be verbs). If the BERT tokenizer breaks a word into multiple tokens, we attach the MEG data to the first token for that word. We align the MEG data with all content words in the fMRI examples (i.e. the content words of the 20 words which precede each fMRI image).

**Cross-validation.**   The fMRI data were recorded in four separate runs in the scanner for each participant. The MEG data were also recorded in four separate runs using the same division of the chapter as fMRI. We cross-validate over the fMRI runs. For each fMRI run, we train the model using the examples from the other three runs and use the fourth run to evaluate the model.

**Preprocessing.**   To preprocess the fMRI data, we exclude the first 20 and final 15 fMRI images from each run to avoid warm-up and boundary effects. Words associated with these excluded images are also not used for MEG predictions. We linearly detrend the fMRI data within run, and standardize the data within run such that the variance of each voxel is 1 and the mean value of each voxel is 0 over the examples in the run. The MEG data is also detrended and standardized within fMRI run (i.e. within cross-validation fold) such that each time-sensor component has mean 0 and variance 1 over all of the content words in the run.

### 3.4 Models and experiments

In this study, we are interested in demonstrating that by fine-tuning a language model to predict brain activity, we can bias the model to encode brain-relevant language information. We also wish to show that the information the model encodes generalizes across multiple experiment participants, and multiple modalities of brain activity recording. For the current work, we compare the models we train to each other only in terms of how well they predict the fMRI data of the nine fMRI experiment participants, but in some cases we use MEG data to bias the model in our experiments. In all of our models, we use a base learning rate of $5 \times 10^{-5}$. The learning rate increases linearly from 0 to $5 \times 10^{-5}$ during the first $10\%$ of the training epochs and then decreases linearly back to 0 during the remaining epochs. We use mean squared error as our loss function in all models. We vary the number of epochs we use for training our models, based primarily on observations of when the models seem to begin to converge or overfit, but we match all of the hyperparameters between two models we are comparing. We also seed random initializations and allocate the same model parameters across our variations so that the initializations are consistent between each pair of models we compare.

**Vanilla model.**   As a baseline, for each experiment participant, we add a linear layer to the pretrained BERT model and train this linear layer to map from the [CLS] token embedding to the fMRI data of that participant. The pretrained model parameters are frozen during this training, so the embeddings do not change. We refer to this model as the vanilla model. This model is trained for either 10, 20, or 30 epochs depending on which model we are comparing this to.

**Participant-transfer model.**   To investigate whether the relationship between text and brain activity learned by a fine-tuned model transfers across experiment participants, we first fine-tune the model on the participant who had the most predictable brain activity. During this fine-tuning, we train only

the linear layer for 2 epochs, followed by 18 epochs of training the entire model. Then, for each other experiment participant, we fix the model parameters, and train a linear layer on top of the model tuned towards the first participant. These linear-only models are trained for 10 epochs, and compared to the vanilla 10 epoch model.

**Fine-tuned model.** To investigate whether a model fine-tuned to predict each participant's data learns something beyond the linear mapping in the vanilla model, we fine-tune a model for each participant. We train only the linear layer of these models for 10 epochs, followed by 20 epochs of training the entire model.

**MEG-transfer model.** We use this model to investigate whether the relationship between text and brain activity learned by a model fine-tuned on MEG data transfers to fMRI data. We first fine-tune this model by training it to predict all eight MEG experiment participants' data (jointly). The MEG training is done by training only the linear output layer for 10 epochs, followed by 20 epochs of training the full model. We then take the MEG fine-tuned model and train it to predict each fMRI experiment participant's data. This training also uses 10 epochs of only training the linear output layer followed by 20 epochs of full fine-tuning.

**Fully joint model.** Finally, we train a model to simultaneously predict all of the MEG experiment participants' data and the fMRI experiment participants' data. We train only the linear output layer of this model for 10 epochs, followed by 50 epochs of training the full model.

**Evaluating model performance for brain prediction using the 20 vs. 20 test.** We evaluate the quality of brain predictions made by a particular model by using the brain prediction in a classification task on held-out data, in a four-fold cross-validation setting. The classification task is to predict which of two sets of words was being read by the participant (Mitchell *et al.*, 2008; Wehbe *et al.*, 2014b,a). We begin by randomly sampling 20 examples from one of the fMRI runs. For each voxel, we take the true voxel values for these 20 examples and concatenate them together – this will be the target for that voxel. Next, we randomly sample a different set of 20 examples from the same fMRI run. We take the true voxel values for these 20 examples and concatenate them together – this will be our distractor. Next we compute the Euclidean distance between the voxel values predicted by a model on the target examples and the true voxel values on the target, and we compute the Euclidean distance between these same predicted voxel values and the true voxel values on the distractor examples. If the distance from the prediction to the target is less than the distance from the prediction to the distractor, then the sample has been accurately classified. We repeat this sampling procedure 1000 times to get an accuracy value for each voxel in the data. We observe that evaluating model performance using proportion of variance explained leads to qualitatively similar results (see Supplementary Figure 4), but we find the classification metric more intuitive and use it throughout the remainder of the paper.

# 4 Results

**Fine-tuned models predict fMRI data better than vanilla BERT.** The first issue we were interested in resolving is whether fine-tuning a language model is any better for predicting brain activity than using regression from the pretrained BERT model. To show that it is, we train the fine-tuned model and compare it to the vanilla model by computing the accuracies of each model on the 20 vs. 20 classification task described in section 3.4. Figure 2 shows the difference in accuracy between the two models, with the difference computed at a varying number of voxels, starting with those that are predicted well by one of the two models and adding in voxels that are less and less well predicted by either. Figure 3 shows where on the brain the predictions differ between the two models, giving strong evidence that areas in the brain associated with language processing are predicted better by the fine-tuned models (Fedorenko and Thompson-Schill, 2014).

**Relationships between text and brain activity generalize across experiment participants.** The next issue we are interested in understanding is whether a model that is fine-tuned on one participant can fit a second participant's brain activity if the model parameters are frozen (so we only do a linear regression from the output embeddings of the fine-tuned model to the brain activity of the second participant). We call this the participant-transfer model. We fine-tune BERT on the experiment participant with the most predictable brain activity, and then compare that model to vanilla BERT.

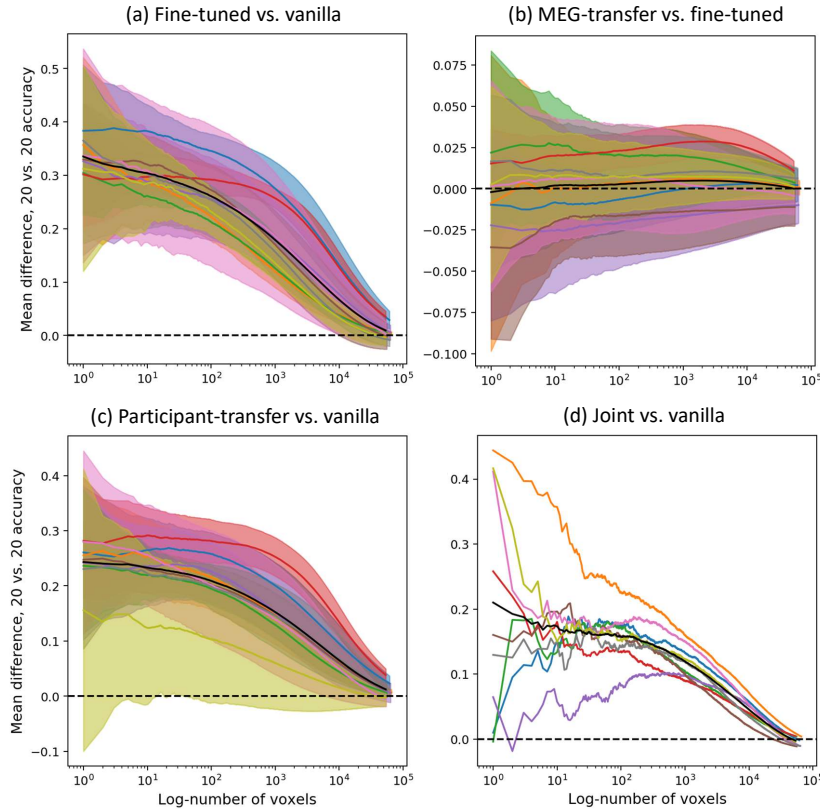

Figure 2: Comparison of accuracies of various models. In each quadrant of the figure above, we compare two models. Voxels are sorted on the x-axis in descending order of the maximum of the two models' accuracies in the 20 vs 20 test (described in section 3.4). The colored lines (one per participant) show differences between the two models' mean accuracies, where the mean is taken over all voxels to the left of each x-coordinate. In (a)-(c) Shaded regions show the standard deviation over 100 model initializations – that computation was not tractable in our framework for (d). The black line is the mean over all participants. In (a), (c), and (d), it is clear that the fine-tuned models are more accurate in predicting voxel activity than the vanilla model for a large number of voxels. In (b), the MEG-transfer model seems to have roughly the same accuracy as a model fine-tuned only on fMRI data, but in figure 3 we see that in language regions the MEG-transfer model appears to be more accurate.

Voxels are predicted more accurately by the participant-transfer model than by the vanilla model (see Figure 2, lower left), indicating that we do get a transfer learning benefit.

**Using MEG data can improve fMRI predictions.** In a third comparison, we investigate whether a model can benefit from both MEG and fMRI data. We begin with the vanilla BERT model, fine-tune it to predict MEG data (we jointly train on eight MEG experiment participants), and then fine-tune the resulting model on fMRI data (separate models for each fMRI experiment participant). We see mixed results from this experiment. For some participants, there is a marginal improvement in prediction accuracy when MEG data is included compared to when it is not, while for others training first on MEG data is worse or makes no difference (see Figure 2, upper right). Figure 3 shows however, that for many of the participants, we see improvements in language areas despite the mean difference in accuracy being small.

**A single model can be used to predict fMRI activity across multiple experiment participants.** We compare the performance of a model trained jointly on all fMRI experiment participants and all MEG experiment participants to vanilla BERT (see Figure 2, lower right). We don't find that this model yet outperforms models trained individually for each participant, but it nonetheless outperforms

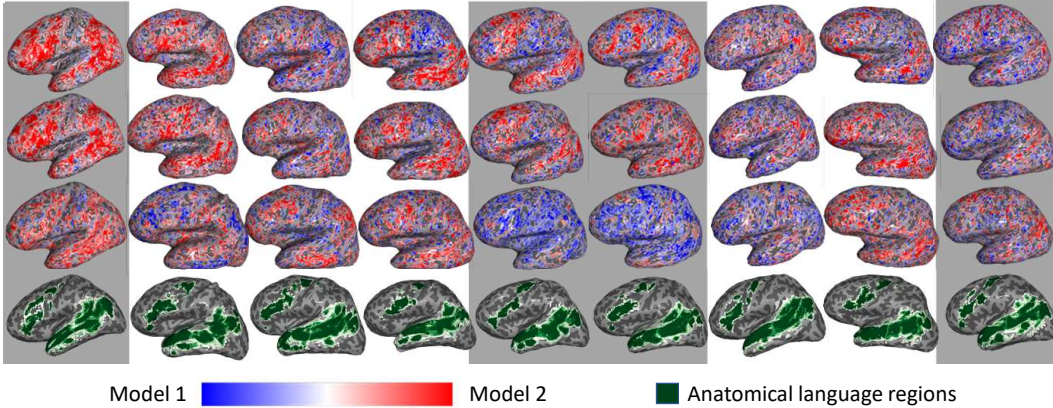

Figure 3: Comparison of accuracies on the 20 vs. 20 classification task (described in section 3.4) at a voxel level for all 9 participants we analyzed. Each column shows the inflated lateral view of the left hemisphere for one experiment participant. Moving from the top to third row, models 1 and 2 are respectively, the vanilla model and the fine-tuned model, the vanilla model and the participant-transfer model, and the fine-tuned model and MEG-transfer model. The leftmost column is the participant on whom the participant-transfer model is trained. Columns with a grey background indicate participants who are common between the fMRI and MEG experiments. Only voxels which were significantly different between the two models according to a related-sample t-test and corrected for false discovery rate at a .01 level using the Benjamini–Hochberg procedure (Benjamini and Hochberg, 1995) are shown. The color-map is set independently for each of the participants and comparisons shown such that the reddest value is at the $95th$ percentile of the absolute value of the significant differences and the bluest value is at the negative of this reddest value. We observe that both the fine-tuned and participant-transfer models outperform the vanilla model, especially in most regions that are considered to be part of the language network. As a reference, we show an approximation of the language network for each participant in the fourth row. These were approximated using an updated version of the Fedorenko *et al.* (2010) language functional parcels[3], corresponding to areas of high overlap of the brain activations of 220 subjects for a "sentences>non-word" contrast. The parcels were transformed using Pycortex (Gao *et al.*, 2015) to each participant's native space. The set of language parcels therefore serve as a strong prior for the location of the language system in each participant. Though the differences are much smaller in the third row than in the first two, we also see better performance in language regions when MEG data is included in the training procedure. Even in participants where performance is worse overall (e.g. the fifth and sixth columns of the third row), voxels where performance improves appear to be systematically distributed according to language processing function. Right hemisphere and medial views are available in the supplementary material.

vanilla BERT. This demonstrates the feasibility of fully joint training and we think that with the right hyperparameters, this model can perform as well as or better than individually trained models.

**NLP tasks are not harmed by fine-tuning.** We run two of our models (the MEG transfer model, and the fully joint model) on the GLUE benchmark (Wang *et al.*, 2018), and compare the results to standard BERT (Devlin *et al.*, 2018) (see Table 1). These models were chosen because we thought they had the best chance of giving us interesting GLUE results, and they were the only two models we ran GLUE on. Apart from the semantic textual similarity (STS-B) task, all of the other tasks are very slightly improved on the development sets after the model has been fine-tuned on brain activity data. The STS-B task results are very slightly worse than the results for standard BERT. The fine-tuning may or may not be helping the model to perform these NLP tasks, but it clearly does not harm performance in these tasks.

**Fine-tuning reduces [CLS] token attention to [SEP] token** We evaluate how the attention in the model changes after fine-tuning on the brain recordings by contrasting the model attention in the fine-tuned and vanilla models described in section 3.4. We focus on the attention from the [CLS] token to other tokens in the sequence because we use the [CLS] token as the pooled output

| Metric | Vanilla | MEG | Joint |
|---|---|---|---|
| CoLA | 57.29 | 57.63 | **57.97** |
| SST-2 | 93.00 | **93.23** | 91.62 |
| MRPC (Acc.) | 83.82 | 83.97 | **84.04** |
| MRPC (F1) | 88.85 | **88.93** | 88.91 |
| STS-B (Pears.) | **89.70** | 89.32 | 88.60 |
| STS-B (Spear.) | **89.37** | 88.87 | 88.23 |
| QQP (Acc.) | 90.72 | **91.06** | 90.87 |
| QQP (F1) | 87.41 | **87.91** | 87.69 |
| MNLI-m | 83.95 | **84.26** | 84.08 |
| MNLI-mm | 84.39 | 84.65 | **85.15** |
| QNLI | 89.04 | **91.73** | 91.49 |
| RTE | 61.01 | **65.42** | 62.02 |
| WNLI | 53.52 | **53.80** | 51.97 |

Table 1: GLUE benchmark results for the GLUE development sets. We compare the results of two of our models to the results published by https://github.com/huggingface/pytorch-pretrained-BERT/ for the pretrained BERT model. The model labeled 'MEG' is the MEG transfer model described in section 3.4. The model labeled 'Joint' is the fully joint model also described in section 3.4. For all but one task, at least one of our two models is marginally better than the pretrained model. These results suggest that fine-tuning does not diminish – and possibly even enhances – the model's ability to perform NLP tasks.

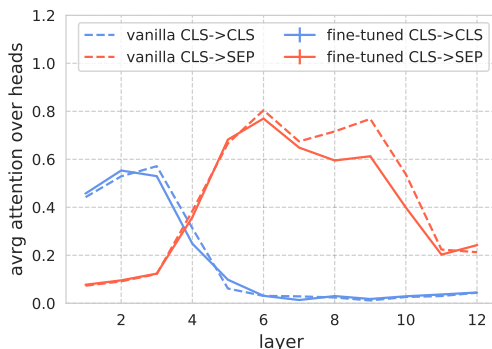

Figure 4: Comparison of attention from the [CLS] token to the [CLS] and [SEP] tokens between vanilla BERT and the fine-tuned BERT (mean and standard error over example presentations, attention heads, and different initialization runs). The attention from the [CLS] token noticeably shifts away from the [SEP] token in layers 8 and 9.

representation of the input sequence. We observe that the [CLS] token from the fine-tuned model puts less attention on the [SEP] token in layers 8 and 9, when compared to the [CLS] token from the vanilla model (see Figure 4). Clark *et al.* (2019) suggest that attention to the [SEP] token in BERT is used as a no-op, when the function of the head is not currently applicable. Our observations that the fine-tuning reduces [CLS] attention to the [SEP] token can be interpreted in these terms. However, further analysis is needed to understand whether this reduction in attention is specifically due to the task of predicting fMRI recordings or generally arises during fine-tuning on any task.

**Fine-tuning may change motion-related representations** In an effort to understand how the representations in BERT change when it is fine-tuned to predict brain activity, we examine the prevalence of various features in the examples where prediction accuracy changes the most after fine-tuning compared to the prevalence of those features in other examples. We score how much the prediction accuracy of each example changes after fine-tuning by looking at the percent change in Euclidean distance between the prediction and the target for our best participant on a set of voxels that we manually select which are likely to be language-related based on spatial location. We average these percent changes over all runs of the model, which gives us 25 samples per example. We take all examples where the absolute value of this average percent change is at least 10% as our set of changed examples, giving us 146 changed examples and leaving 1022 unchanged examples. We then compute the probability that each feature of interest appears on a word in a changed example and compare this to the probability that the feature appears on a word in an unchanged example, using bootstrap resampling on the examples with 100 bootstrap-samples to estimate a standard error on these probabilities. The features we evaluate come from judgments done by Wehbe *et al.* (2014b) and are available online[4]. The sample sizes are relatively small in this analysis and should be viewed as preliminary, however, we see that examples containing verbs describing movement and imperative language are more prevalent in examples where accuracies change during fine-tuning. See the supplementary material for further discussion and plots of the analysis.

# 5 Discussion

This study aimed to show that it is possible to learn generalizable relationships between text and brain activity by fine-tuning a language model to predict brain activity. We believe that our results provide several lines of evidence that this hypothesis holds.

First, because a model which is fine-tuned to predict brain activity tends to have higher accuracy than a model which just computes a regression between standard contextualized-word embeddings and brain activity, the fine-tuning must be changing something about how the model encodes language to improve this prediction accuracy.

Second, because the embeddings produced by a model fine-tuned on one experiment participant better fit a second participant's brain activity than the embeddings from the vanilla model (as evidenced by our participant-transfer experiment), the changes the model makes to how it encodes language during fine-tuning at least partially generalize to new participants.

Third, for some participants, when a model is fine-tuned on MEG data, the resulting changes to the language-encoding that the model uses benefit subsequent training on fMRI data compared to starting with a vanilla language model. This suggests that the changes to the language representations induced by the MEG data are not entirely imaging modality-specific, and that indeed the model is learning the relationship between language and brain activity as opposed to the relationship between language and a brain activity *recording modality*.

Models which have been fine-tuned to predict brain activity are no worse at NLP tasks than the vanilla BERT model, which suggests that the changes made to how language is represented improve a model's ability to predict brain activity without doing harm to how well the representations work for language processing itself. We suggest that this is evidence that the model is learning to encode brain-activity-relevant language information, i.e. that this biases the model to learn representations which are better correlated to the representations used by people. It is non-trivial to understand exactly how the representations the model uses are modified, but we investigate this by examining how the model's attention mechanism changes, and by looking at which language features are more likely to appear on examples that are better predicted after fine-tuning. We believe that a more thorough investigation into how model representations change when biased by brain activity is a very exciting direction for future work.

Finally, we show that a model which is jointly trained to predict MEG data from multiple experiment participants and fMRI data from multiple experiment participants can more accurately predict fMRI data for those participants than a linear regression from a vanilla language model. This demonstrates that a single model can make predictions for all experiment participants – further evidence that the changes to the language representations learned by the fine-tuned model are generalizable. There are optimization issues that remain unsolved in jointly training a model, but we believe that ultimately it will be a better model for predicting brain activity than models trained on a single experiment participant or trained in sequence on multiple participants.

# 6 Conclusion

Fine-tuning language models to predict brain activity is a new paradigm in learning about human language processing. The technique is very adaptable. Because it relies on encoding information from targets of a prediction task into the model parameters, the same model can be applied to prediction tasks with different sizes and with varying temporal and spatial resolution. Additionally it provides an elegant way to leverage massive data sets in the study of human language processing. To be sure, more research needs to be done on how best to optimize these models to take advantage of multiple sources of information about language processing in the brain and on improving training methods for the low signal-to-noise-ratio setting of brain activity recordings. Nonetheless, this study demonstrates the feasibility of biasing language models to learn relationships between text and brain activity. We believe that this presents an exciting opportunity for researchers who are interested in understanding more about human language processing, and that the methodology opens new and interesting avenues of exploration.

**Acknowledgments**

This work is supported in part by National Institutes of Health grant no. U01NS098969 and in part by the National Science Foundation Graduate Research Fellowship under Grant No. DGE1745016.

## Footnotes

[1] http://www.cs.cmu.edu/~fmri/plosone/

[2] https://github.com/huggingface/pytorch-pretrained-BERT/

[3]https://evlab.mit.edu/funcloc/download-parcels

[4]http://www.cs.cmu.edu/~fmri/plosone/

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
