[Supplementary Material · BrainBert_supplementary.pdf]

# Inducing brain-relevant bias
# in natural language processing models,
# supplementary material

**Dan Schwartz**
Carnegie Mellon University
drschwar@cs.cmu.edu

**Mariya Toneva**
Carnegie Mellon University
mariya@cmu.edu

**Leila Wehbe**
Carnegie Mellon University
lwehbe@andrew.cmu.edu

## Abstract

This supplementary material is for the paper "Inducing brain-relevant bias in natural language processing models" from the proceedings of NeurIPS 2019, which describes fine-tuning BERT, a recently introduced language model, to predict brain activity. The supplementary material includes additional views of each experiment participant's brain in the voxel-level comparisons of fMRI prediction accuracies across differently trained models, a comparison of different models in terms or proportion of variance explained rather than the 20 vs. 20 accuracy from the main paper, and plots of the prevalence of various story features in examples that change the most during fine-tuning.

## 1    Additional views of voxel-level comparisons

In the main paper, Figure 3 in the results section shows a summary of spatial distributions of changes in fMRI prediction accuracy after fine-tuning by showing lateral views of the left hemisphere of all nine experiment participants across three different models. In Supplementary Figures 1, 2, and 3 we break out the three different models into separate figures and include the right hemisphere and medial views for each participant.

## 2    Model comparison using proportion of variance explained

Although we believe that the 20 vs. 20 accuracy described in the main paper (Mitchell *et al.*, 2008; Wehbe *et al.*, 2014b,a) gives a more intuitive comparison of models than the proportion of variance explained, both metrics have value. In some ways the proportion of variance explained is more sensitive to changes since the accuracy quantizes the results. Supplementary Figure 4 shows the same results as Figure 2 from the main paper, but in terms of proportion of variance explained rather than 20 vs. 20 accuracy. The results are qualitatively similar, but we can even more clearly see the effects of overfitting in the models as the proportion of variance explained becomes negative when we include all voxels in the mean difference.

## 3    Prevalence of story features in the most changed examples

In an effort to understand how the representations in BERT change when it is fine-tuned to predict brain activity, we examine the prevalence of various features in the examples where prediction accuracy changes the most after fine-tuning compared to the prevalence of those features in other examples. We score how much the prediction accuracy of each example changes after fine-tuning by looking at the percent change in Euclidean distance between the prediction and the target for our

Figure 1: Comparison of accuracies on the 20 vs. 20 classification task between the fine-tuned and vanilla models (described in the main paper) at a voxel level for all 9 participants we analyzed. Moving from left to right across the page, the columns show inflated lateral views of the right and left hemisphere followed by inflated medial views of the left and right hemisphere respectively. Each row shows one participant. Only voxels which were significantly different between the two models according to a related-sample t-test and corrected for false discovery rate at a .01 level using the Benjamini–Hochberg procedure (Benjamini and Hochberg, 1995) are shown. The color-map is set independently for each of the participants such that the reddest value is at the $95th$ percentile of the absolute value of the significant differences and the bluest value is at the negative of this reddest value. The fine-tuned model tends to have higher prediction accuracy than the vanilla model in language areas.

Figure 2: Comparison of accuracies on the 20 vs. 20 classification task between the participant-transfer and vanilla models (described in the main paper) at a voxel level for all 9 participants we analyzed. Moving from left to right across the page, the columns show inflated lateral views of the right and left hemisphere followed by inflated medial views of the left and right hemisphere respectively. Each row shows one participant. Only voxels which were significantly different between the two models according to a related-sample t-test and corrected for false discovery rate at a .01 level using the Benjamini–Hochberg procedure (Benjamini and Hochberg, 1995) are shown. The color-map is set independently for each of the participants such that the reddest value is at the $95th$ percentile of the absolute value of the significant differences and the bluest value is at the negative of this reddest value. Like the fine-tuned model, the participant-transfer model tends to have higher prediction accuracy than the vanilla model in language areas.

Figure 3: Comparison of accuracies on the 20 vs. 20 classification task between the MEG-transfer and fMRI only models (fMRI only is referred to elsewhere as the fine-tuned model, both models are described in the main paper) at a voxel level for all 9 participants we analyzed. Moving from left to right across the page, the columns show inflated lateral views of the right and left hemisphere followed by inflated medial views of the left and right hemisphere respectively. Each row shows one participant. Rows with a grey background indicate participants whose data were used in both the MEG training and the fMRI training (i.e., these participants were scanned in separate sessions in both modalities). Only voxels which were significantly different between the two models according to a related-sample t-test and corrected for false discovery rate at a .01 level using the Benjamini–Hochberg procedure (Benjamini and Hochberg, 1995) are shown. The color-map is set independently for each of the participants such that the reddest value is at the $95th$ percentile of the absolute value of the significant differences and the bluest value is at the negative of this reddest value. For most participants, prediction of language areas improves with the MEG-transfer model. However, the results are mixed and for some participants, prediction in (some) language areas is arguably worse. Nonetheless, the results suggest that the changes to the model learned during training to predict MEG are helpful for predicting fMRI data in language areas.

Figure 4: Comparison of accuracies of various models. In each quadrant of the figure above, we compare two models. Voxels are sorted on the x-axis in descending order of the maximum of the two models' proportion of variance explained. The colored lines (one per participant) show differences between the two models' mean proportion of variance explained, where the mean is taken over all voxels to the left of each x-coordinate. In (a)-(c) Shaded regions show the standard deviation over 100 model initializations – that computation was not tractable in our framework for (d). The black line is the mean over all participants. In (a), (c), and (d), it is clear that the fine-tuned models are more accurate in the prediction of voxels than the vanilla model for a large number of the voxels. In (b), the results are more mixed. The MEG-transfer model seems to have roughly the same proportion of variance explained as a model fine-tuned only on fMRI data, but in Figure 3 we see that in language regions, for most participants, the MEG-transfer model appears to be more accurate.

Figure 5: Voxels used to compute changes in accuracy between the fine-tuned and vanilla models for the feature distribution analysis described in Section 3. From left to right in the figure, are inflated lateral views of the right and left hemispheres followed by inflated medial views of the left and right hemispheres. Voxel selections are done manually based on location in the brain with the goal of restricting the accuracy computation to areas that are more likely to be involved in language processing.

Figure 6: Prevalence of the motion-related labels on words in examples that are most and least changed in terms of prediction accuracy in language areas. The set of examples that is most and least changed is computed as described in Section 3. The most dramatic change in feature distribution among all features we examine, motion or otherwise, is the 'move' label.

best participant on a set of voxels that we manually select which are likely to be language-related based on spatial location (see Figure 5). We average these percent changes over all runs of the model, which gives us 25 samples per example. We take all examples where the absolute value of this average percent change is at least $10\%$ as our set of changed examples, giving us 146 changed examples and leaving 1022 unchanged examples. We then compute the probability that each feature of interest appears on a word in a changed example and compare this to the probability that the feature appears on a word in an unchanged example, using bootstrap resampling on the examples with 100 bootstrap-samples to estimate a standard error on these probabilities. The features we evaluate come from judgments done by Wehbe *et al.* (2014b) and are available online[1]. We examine all available features, but here we present only motion labels (Figure 6), emotion labels (Figure 7), and part-of-speech labels (Figure 8), as other features were either too sparse to evaluate or did not show any change in distribution. The sample sizes are relatively small in this analysis and should be viewed as preliminary, however, we see that examples containing verbs describing movement and imperative language are more prevalent in examples where accuracies change during fine-tuning. We believe the method of fine-tuning a model and evaluating feature distributions among the most changed examples is an exciting direction for future work.

Figure 7: Prevalence of the emotion-related labels on words in examples that are most and least changed in terms of prediction accuracy in language areas. The set of examples that is most and least changed is computed as described in Section 3. There is some indication that representations for imperative language is changed during fine-tuning, as indicated by the change in prevalence of the 'commanding' label, but note that the prevalence is low for both the most changed and least changed examples, so this could easily be a sampling effect.

Figure 8: Prevalence of the part-of-speech labels on words in examples that are most and least changed in terms of prediction accuracy in language areas. The set of examples that is most and least changed is computed as described in Section 3. Verbs and determiners seem to be candidates for further study.

## Footnotes

[1]`http://www.cs.cmu.edu/~fmri/plosone/`