[Reviews · NeurIPS 2019]

Reviewer 1



Originality: The underlying experimental framework is novel and thorough. The fine tuning of BERT is a small but attractive idea. The overall study is clearly novel and the empirical results are also new and significant. Quality: The paper is technically sound. Clarity: The paper is well written. The study is logically and systematically laid out in an easy to read manner. Significance: This does advance the state of the art reasonably.

Reviewer 2



Notwithstanding the significant contributions of the paper discussed above, there is perhaps too much focus on overall performance metrics and too little scientific investigation of how BERT and brain data relate to each other. How do representations in the pre-trained BERT model and the brain-fine-tuned BERT model encode linguistic information, and what is gained by fine-tuning? It would be interesting to compare the two models to try and understand what brain-relevant linguistic information is absent from the original BERT instantiation. It is arguably a bit tautological to argue that “fine-tuning models to predict recordings of brain activity … will lead to representations that encode more brain-activity relevant language information” (Abstract lines 6-7; Also lines 143-144). Isn’t the more interesting question about what commonalities exist between brain representations of linguistic information and BERT representations of linguistic information? What kind of language information is “brain-activity relevant language information”? For example, emotional sentiment may be a particularly important aspect of the brain data, but not a very salient component of the representations in BERT, though sentiment may be decodable from BERTs embeddings (and thus can be magnified by the brain fine tuning process). Lines 120-121: 20 words are used as the relevant context for each fMRI image, but these 20 words do not respect sentence boundaries. Does this mean that the input to BERT during fine-tuning does not respect sentence boundaries? If so, this seems undesirable - - it introduces a discrepancy between the format of the input in the initial BERT training and the format of the input in the current fine-tuning. Therefore, we cannot be sure whether changes to BERTs linguistic representations as a result of fine-tuning are a result of the neurocogntive signal relating to the input, or to the new input scheme (e.g. when it comes to the GLUE evaluations).

Reviewer 3



This paper addresses a young and exciting area of research linking cognitive neuroscience and artificial intelligence. While the methods are mostly reasonable, I find the paper very lacking in both framing and interpretation of results. ## Significance I don’t think this study is well motivated anywhere in the paper. Why is it of interest whether or not a natural language processing model can accurately model brain activations across participants? What do we expect to learn (about either brains or models) by finding the answer to that question? Is it reasonable to think that the results could have been otherwise — that, for example, fine-tuning on predicting brain activations would have made the model worse at predicting brain activations (within-subject or across-subject)? The introduction suggests that this paper might be of interest re: combining information from multiple neuroimaging modalities. But the simple co-training method is not particularly interesting in this respect — for contrast, compare with e.g. other computational models for fusing multimodal data [see e.g. 2]. ## Quality ### Interpretation The paper does not attempt to provide any explanation about why prediction performance changes the way it does. The most substantial analysis I could find was on p. 7: “the fine-tuning must be changing something about how the model encodes language to improve this prediction accuracy.” The results further show that this “something” is at least partially shared between imaging modalities. What is it about predicting brain activation data, then, that isn’t already present in the pre-trained BERT model? If the framing of this research has to do with learning about either artificial neural network models or human language processing, then it’d be good to have an answer to this question. ### Evaluation It is very unclear how to interpret the NLP evaluation results in Table 1. Most of the quantitative changes are very small. What would the difference in performance between N random restarts of the BERT model look like in this table? It might be interesting/useful to run a statistical evaluation (pairwise t-test or sign test, depending on the data) to better understand the changes between the vanilla and fine-tuned models. 20 vs. 20 is a very coarse evaluation measure and not well justified a priori — this will seem strange to audiences less familiar with brain encoding/decoding. Please explain why it is a reasonable measure for the model selection you are doing in this paper (see e.g. [3] p238 left column). ## Clarity Figure 2 is quite difficult to interpret at a glance — it would be useful to have a higher-level summary figure of some sort. ## Originality I believe this work is original, though its methods largely overlap with those of [1]. It is a valid separate line of work from the more common brain encoding/decoding papers, which don’t investigate fine-tuning. ## References [1] Schwartz, D., & Mitchell, T. (2019). Understanding language-elicited EEG data by predicting it from a fine-tuned language model. arXiv preprint arXiv:1904.01548. [2] Calhoun, V., Adali, T., & Liu, J. (2006, August). A feature-based approach to combine functional MRI, structural MRI and EEG brain imaging data. In 2006 International Conference of the IEEE Engineering in Medicine and Biology Society (pp. 3672-3675). IEEE. [3] Wehbe, L., Vaswani, A., Knight, K., & Mitchell, T. (2014, October). Aligning context-based statistical models of language with brain activity during reading. In Proceedings of the 2014 Conference on Empirical Methods in Natural Language Processing (EMNLP) (pp. 233-243).

Reviewer 4



This is high-quality work that is certainly mature enough for presentation at NeuroIPS. It's original and well-written. In particular, I appreciated Figure 3. It's a compelling visualization of the fact that the fine-tuning is doing something relevant in brain regions that have been associated with language.

[Author Response · NeurIPS 2019]

We thank the reviewers for the thoughtful suggestions and attempt to address their questions within the space constraints.

**All reviewers:** To better interpret our results, we have a new analysis using additional labels released with the Harry Potter data which identify the presence of various syntactic, semantic, and emotional features for each word in the chapter. We score each input example of 20 words as to how much fine-tuning hurts or harms the example. On manually selected language-region voxels, we compute the difference in the distance from the model prediction to the target between the fine-tuned and vanilla BERT models (for our best participant). We compare the distributions of the features on the examples most helped and most harmed by fine-tuning, as determined by this metric, and find some indications that features related to emotion, the subject dependency-role, and noun representations are improved by fine-tuning. We will present this analysis in the main paper, and we think that better understanding the changes in the model will be an exciting area for future research.

**R2 and R3:** Our use of the 20 vs. 20 evaluation follows previous work using this dataset (cf. Wehbe *et al.* 2014a, 2014b in the paper). In this experiment the text is shown to the participant only once, so the SNR is very low. 20 vs. 20 boosts the SNR without the averaging that is normally used in a multiple repetition setting. It enables us to compare models more easily than $R^2$ which is dominated by noise. Qualitatively the brain maps of prediction performance in our model comparisons look similar using either metric, and we will add the $R^2$ maps as a supplementary figure.

We agree that we needed to quantify the results in fig. 2. For the models where it was computationally feasible (all but the fully jointly trained model) we trained the models 100 times ($25 \times \langle 4 \text{ CV folds} \rangle$) with different initializations. The models all use the same initialization for run $i$ so we use a paired t-test per voxel to evaluate whether voxel prediction accuracies are different between two models, correcting for false discovery rate at a .01 level (Benjamini 1995 JRoyalStatSoc). We plan to replace fig. 2 in the main paper with fig. 4 (a-c) for the models where we can, and to replace fig. 3 in the main paper with statistical maps similar to fig. 4 (f-i) here. Fig. 4 (f-i) shows that while the MEG to fMRI transfer learning does not appear to improve voxel prediction on average (fig. 4 (c), for almost every participant it helps prediction in language regions and harms prediction elsewhere (compare (f-i) with (d)). The harm is likely due to overfitting.

Figure 4: **(a-c):** Each sub-figure compares two models. Voxels are sorted on the x-axis in descending order of the maximum of the two models' accuracies. The colored lines (one per participant) show mean accuracies taken over all voxels to the left of each x-coord. Shaded regions show std. deviation over 100 model initializations. The black line is the mean over all participants. **(d):** Regions typically associated with language processing (adapted from Fedorenko *et al.* 2012 Neuropsychologia). **(e):** An example significance mask. **(f-i):** Comparison of the MEG-transfer and fine-tuned models (left hemispheres). Voxels are masked out where differences are not significant and colored according to the mean difference in accuracy between the MEG transfer and fine-tuned models. Notice the bottom parts of plots (e-f) look different than (d) because they don't include the cerebellum.

This outcome also relates to motivation. It is not self-evident that fine-tuning should help prediction. It is possible that there could be nothing to learn beyond what is encoded in BERT (i.e. for vanilla to be the best possible fit even in a setting with a large number of samples), or for overfitting to be too problematic in a practical setting (with a small number of samples). We demonstrate that prediction of language areas improves while prediction elsewhere does not.

**R1 and R4:** Thank you for the positive evaluation. We will add more detail about the CLS token as per R4's suggestion.

**R2:** When we run GLUE, we use normal inputs (i.e. we do not use 20-word inputs). A bidirectional model is only problematic if we are attempting to model how information transformations happen in the brain algorithmically. Here we are interested in nudging the information content in BERT to be similar to the brain, but not in making its algorithm similar to the brain. Since humans have more real word knowledge from which to make predictions as they read left-to-right — helping in tasks like anaphora resolution — it's plausible for a bidirectional model to be more similar to human representations by using right-to-left processing to make up for its lack of knowledge.

**R3:** We agree this paper was light on framing, we have attempted to provide some backing in the short space here but we will appropriately motivate the problem in the main paper. Please see above for suggested interpretation. Clarification: we do not claim to be improving on the vanilla BERT w.r.t. GLUE, only that we are not impairing performance.

[Meta-Review · NeurIPS 2019]

The reviewers agree that this paper makes an original contribution in investigating the fine tuning of a contextual embedding model, e.g. BERT, on neurological data. They have identified some issues with the clarity of the motivation of the work and the presentation of some of the results, but I feel that these shortcomings are outweighed by the contributions of this work which should be of interest to researchers across a number of disciplines. The reviewers thank the authors for the additional analysis provided in their response and look forward to this being incorporated into the final paper. In particular it would be good to clarify to significance of the results in Table 1, whether any of the differences are interpretable or whether this simply shows that they are equivalent.